# Antibiotics Resistance and Virulence of *Staphylococcus aureus* Isolates Isolated from Raw Milk from Handmade Dairy Retail Stores in Hefei City, China

**DOI:** 10.3390/foods11152185

**Published:** 2022-07-22

**Authors:** Hui Wang, Jiawei Shen, Chengfeng Zhu, Kai Ma, Mengcheng Fang, Bingbing Li, Wenhui Wang, Ting Xue

**Affiliations:** 1School of Life Sciences, Anhui Agricultural University, Hefei 230036, China; wang28hui@163.com (H.W.); shenjwshirley@outlook.com (J.S.); zhucf@ahau.edu.cn (C.Z.); makaiunique@163.com (K.M.); f17356533998@163.com (M.F.); bingbingli@ahau.edu.cn (B.L.); wangwenhui@ahau.edu.cn (W.W.); 2Food Procession Research Institute, Anhui Agricultural University, Hefei 230036, China

**Keywords:** handmade dairy products, raw milk, *Staphylococcus aureus*, prevalence and characteristics

## Abstract

Handmade dairy products, which retain the nutrients in milk to the greatest extent, have become popular in China recently. However, no investigation regarding the characteristics of *Staphylococcus aureus* (*S. aureus*) in raw milk of handmade dairy retail stores has been reported. Here, we investigated the antimicrobial susceptibility, virulence, biofilm formation, and genetic diversity of *S. aureus* in raw milk from handmade dairy retail stores in Hefei, China. After 10 months of long-term monitoring, 50 *S. aureus* strains were isolated from 69 different raw milk samples, of which 6 were positive for methicillin-resistant *S. aureus* (MRSA). The resistance rates of these isolates to ampicillin, erythromycin, kanamycin, tetracycline, sulfamethoxazole-trimethoprim, gentamicin, ofloxacin, oxacillin, chloramphenicol, and doxycycline were 56, 54, 40, 24, 22, 22, 18, 14, 8 and 6%, respectively. All 50 isolates were susceptible to vancomycin and 29 strains (58%) showed multidrug resistance phenotype. For enterotoxins genes, *selp* (14%) was detected the most frequently, followed by *sea* (6%), *sec* (4%), *sei* (4%), *ser* (4%), *selj* (4%), and *seh* (2%). By microplate assay, 32 and 68% of the strains showed moderate and strong biofilm formation ability, respectively. Fifty isolates were discriminated into nine *spa* types, and the most common *spa* typing was t034 (42%). The results of this study indicate that *S. aureus* from raw milk may constitute a risk concerning food poisoning, and more attention must be given to awareness and hygienic measures in the food industry.

## 1. Introduction

*Staphylococcus aureus* (*S. aureus*) is a kind of Gram-positive opportunistic pathogenic bacterium, which can cause a variety of zoonotic infections and toxin-type food poisoning [1]. The outbreaks of staphylococcal food poisoning (SFP) caused by *S. aureus* and its enterotoxins have been reported worldwide [2,3]. The Center for Disease Control and Prevention’s assessment of foodborne illness from 2006 to 2008 shows that there are 241,188 cases of SFP in the United States every year, resulting in 1064 hospitalizations and 6 deaths [4]. In China, microbes accounted for 53.7% of food poisoning cases in 2015. *S. aureus* is one of the most important pathogenic bacteria [5]. The onset of SFP is rapid, which makes patients feel nauseous, vomit, and experience abdominal cramps [2].

Dairy products account for a certain proportion of all reported SFP cases [6]. Milk and dairy products are favored by consumers as an essential source of protein, calcium, and vitamins [7,8]. Of these, milk powder, rich in micronutrients, is used worldwide as the main ingredient in infant formula [9]. Although dairy products have experienced a series of treatments such as high temperature, high pressure, and drying, there are still many reports of *S. aureus* detected in dairy products [10,11]. Currently, handmade retail products, such as handmade yogurt and cheese, are becoming more and more popular, but the way they are handled may be inadequate [12]. There may be a potential hazard of incomplete sterilization of food. Simultaneously, the source of raw milk is another important aspect to ensure the safety of final products. Residues of *S. aureus* in dairy products may pose a risk of food poisoning. Consequently, it is necessary to investigate the characteristics of *S. aureus* in the raw milk from handmade dairy retail stores.

*S. aureus* enterotoxins (SEs) are gastrointestinal exotoxins [13]. The SEs remain active in the digestive tract after being ingested by humans because they can resist proteolytic enzymes at high temperatures [6]. Previous studies have shown that SEs are considered to be the major cause of SFP [14]. Enterotoxin contains the classic SEs encoded by *sea*, *seb*, *sec*, *sed*, and *see* genes, novel enterotoxin encoded by *seg*, *seh*, *sei*, and *ser* genes, and enterotoxin-like (SE*l*s) protein encoded by *selj* and *selp* genes [6,15]. Due to antibiotic abuse, the resistance of *S. aureus* is gradually increasing, and different regions show different epidemic trends. It was previously reported that *S. aureus* with antibiotic resistance has caused foodborne disease outbreaks [16,17], especially the methicillin-resistant *S. aureus* (MRSA) and multidrug resistance (MDR) *S. aureus*, posing a public health security challenge [18]. Biofilm substrates not only help bacteria store nutrients and water but also reduce the rate of antibiotic penetration [19,20]. Moreover, in the pathogenic process of *S. aureus,* biofilm is a crucial virulence factor, which helps bacteria survival in harsh environments [21,22,23,24]. As a result, it is important to monitor the antibiotics resistance trend and virulence-associated SEs gene and biofilm formation rate of *S. aureus* in the raw milk from handmade dairy retail stores.

The *spa* typing is a genotyping method based on DNA sequence, which is specially used for the typing of *S. aureus* [25]. The *spa* typing based on repeats in the staphylococcal protein A sequence allows higher discrimination than does multilocus sequence typing and it remains a popular typing method [26]. The *spa* gene can be genotyped by polymerase chain reaction (PCR) and DNA sequencing. Some studies have found that specific *S. aureus* lineages may be geographically prevalent and exhibit specific patterns of antibiotic resistance and virulence [27,28]. Accordingly, a better understanding of the genotypes of *S. aureus* isolates from raw milk may bring in more effective measures to reduce the occurrence of SFP and trace the source of infection [29]. The recent hotspot studies aimed to investigate the prevalence and molecular characteristics of *S. aureus* isolate from raw milk from dairy farms [27,30]. Unfortunately, there have been no reports on the prevalence and long-term detection of *S. aureus* in raw milk from handmade dairy products.

In this study, we conducted a ten-month detection of *S. aureus* in raw milk from handmade dairy products retail stores in Hefei city, China. The epidemic characteristics of *S. aureus* through strain isolation and identification, *spa* typing, drug resistance measurement, biofilm formation assay in vitro, and enterotoxin encoding genes detection were analyzed.

## 2. Materials and Methods

### 2.1. Collection of Raw Milk Samples

From August 2020 to May 2021, a total of 69 different raw milk samples were collected from handmade dairy products retail stores in Hefei, Anhui Province, China. The collected 50 mL raw milk was cryogenically refrigerated in a sterile tube and transported to the laboratory within 1 h for subsequent studies.

### 2.2. Identification and Isolation of S. aureus

In total, 50 mL raw milk was poured into 50 mL tryptic soy broth (TSB; Difco) with 7.5% NaCl and incubated for 16 h at 37 °C with continuous shaking. The cultures were spread on tryptic soy agar (TSA; Difco, Franklin Lakes, NJ, USA) plates cultivated for 16 h at 37 °C. A single colony was then cultivated overnight in 3 mL TSB. Genomic DNA of every isolate was extracted using TIANamp Bacteria DNA Kit (TianGen Biotech, Beijing, China), respectively, and 4 µg/mL lysostaphin was added if necessary. These strains were identified as *S. aureus* by 16S rDNA gene sequencing and PCR analysis of the thermonuclease (*nuc*) gene-specificity [31]. The confirmed *S. aureus* isolates were kept at −80 °C in 25% glycerol for further study. All primers used in this study are listed in Table 1.

### 2.3. Detection of mecA and Genes Encoding Enterotoxin

All *S. aureus* isolates were screened for *mecA* and genes encoding enterotoxin by PCR amplification using primers in Table 1. The primers were synthesized by Tsingke Biotechnology Co., Ltd. (Nanjing, China). The amplified *mecA*-gene-positive strain was defined as MRSA strain [32], with *mecA*-positive *S. aureus* N315 as the positive control [33]. Several genes encoding associated SEs were tested by PCR, including *sea, seb, sec, sed, see, ser, seg, seh, sei, selj,* and *selp* [15]. The SEs genes were amplified by PCR at 94 °C for 4 min, followed by 30 cycles of 94 °C for 30 s, 55 °C for 30 s, and 72 °C for 40 s, and final extension for 10 min at 72 °C.

### 2.4. Antibiotic Susceptibility Testing

Disk diffusion was conducted to test the antimicrobial susceptibility of all isolates in accordance with the guidelines of the Clinical and Laboratory Standards Institute (CLSI, 2015) [27]. Ampicillin (10 μg), gentamicin (10 μg), kanamycin (30 μg), tetracycline (30 μg), doxycycline (30 μg), sulfamethoxazole-trimethoprim (1.25/23.75 µg), chloramphenicol (30 μg), erythromycin (15 μg), and ofloxacin (5 μg) were used as antimicrobial agents (Oxoid, Basingstoke, UK). The sensitivity of vancomycin and oxacillin was tested by broth microdilution according to the guidelines of CLSI (CLSI, 2015). *S. aureus* ATCC25923 and ATCC29213 were used as quality control strains, and the testing experiment was repeated twice.

### 2.5. Biofilm Formation Assay

The method of biofilm quantification was performed as described previously and modified herein [34]. Briefly, all isolates were grown in TSB for 16 h; *S. aureus* from the overnight growth was diluted to an optical density at 600 nm (OD_600_) of around 0.03 in fresh TSB for the following subsequent assays. After being cultivated for 4 h at 37 °C with shaking at 180 rpm, the cultures were diluted with fresh TSB to an optical density at 600 nm (OD_600_) of about 1.00 and diluted at 1:100 with fresh TSB. The cultures were then transferred into sterile 96-well flat-bottomed tissue culture plates for the following incubation at 37 °C for 24 h. The adherent bacteria were stained with 0.2% crystal violet for 30 min and then washed three times with sterile phosphate-buffered saline (PBS). The biomass of the biofilm was dissolved with 33% acetic acid and then determined quantitatively by using a Micro ELISA auto-reader at the wavelength of 492 nm. For the biofilm production assays, *S. aureus* NCTC8325, a strong biofilm-former, was regarded as the positive control [35], and sterile TSB was selected as the negative control. An OD_492nm_ value of 0.6 was applied as the cutoff point to distinguish between biofilm-formers and non-biofilm-formers (cutoff (ODc) = average OD + SD of 3 negative control) [36]. Biofilm formation was classified as strong (OD492 > 1.71), moderate (1.71 > OD492 > 0.6), and weak (OD492 < 0.6) [37].

### 2.6. spa Typing

The *spa* typing was performed by amplification of polymorphic X region of the *S. aureus* protein A gene (*spa*), using the standard primers spa-1113F (5′-TAAAGACGA-TCCTTCGGTGAGC-3′) and spa-1514R (5′-CAGCAGTAGTGCCGTTTGCTT-3′). The primers and protocol are available on the Ridom Spa Server database (http://www.spaserver.ridom.de, accessed on 3 October 2021). It was conducted according to methods described previously and modified as described herein [38]. Briefly, the PCR reaction mix contained 250 nmol of each primer, 12.5 µL of 2× PrimeSTAR^®^ Max DNA Polymerase (Takara Bio Inc., Dalian, China), and 1 µL of DNA template, and genomic DNA was extracted according to the manufacturer’s TIANamp Bacteria DNA Kit instructions (TianGen Biotech, Beijing, China). The PCR was performed under the following conditions: initial denaturation at 98 °C for 5 min, 32 cycles at 98 °C for 10 s, 60 °C for 15 s, and 72 °C for 30 s, and a final extension of 10 min at 72 °C. All the PCR products were sequenced by Tsingke Biotechnology Co., Ltd. (Nanjing, China), and then *spa* type was identified using this database.

### 2.7. Statistical Analysis

Statistical analyses were performed using SPSS software (SPSS standard, version 18.0; *SPSS*, Inc., Chicago, IL, USA) to analyze the differences in the prevalence, antimicrobial resistance, distribution of virulence or enterotoxin-producing genes, and biofilm formation ability. A *p* < 0.05 was considered statistically significant.

## 3. Results

### 3.1. Isolation and Identification of S. aureus

A total of 69 consecutive and non-repetitive raw milk samples were collected during the 10-month monitoring of raw milk from artisanal dairy retail stores in downtown Hefei, China. A total of 50 *S. aureus* isolates were identified by 16S rDNA gene sequencing as well as by PCR analysis of the *nuc* gene specific to this species, and the detection rate of *S. aureus* in raw milk samples was 72.5%. Six of the *S. aureus* isolates harbored *mecA* gene and were identified as MRSA strains (Table 2, Figure 1).

### 3.2. spa Typing

The *spa* typing information of the 50 isolates is shown in Table 3 and Figure 1. They were divided into 9 *spa* types. The most prevalent *spa* type was t034 (42.0%, 21/50). The other 8 *spa* types were: t3904, t189, t4431, t030, t527, t2844, t267, and t4682, and their proportions were 14, 8, 10, 6, 8, 4, 4 and 4%, respectively.

### 3.3. Distribution of Enterotoxin Genes

The production of enterotoxin is a potential factor causing SPF. Eleven enterotoxin genes (including *sea*, *seb*, *sec*, *sed*, *see*, *seg*, *seh*, *sei*, *ser*, *selj*, and *selp*) were selected to test the potential of the *S. aureus* isolates to produce enterotoxin. Results showed that 7 enterotoxin genes were detected in the 50 *S. aureus* isolates (Table 4 and Figure 1). The enterotoxin genes *seb, see, seg,* and *sed* were not found in any isolate. As shown in Table 4, the 7 enterotoxin genes, *selp, sea, sec, sei, ser, selj*, and *seh* were detected in 7 (14%), 3 (6%), 2 (4%), 2 (4%), 2 (4%), 2 (4%), and 1 (2%) isolates, respectively. In total, these enterotoxin genes were identified in 24% (12/50) of the *S. aureus* isolates, and 6% (3/50) of the isolates contained 3 enterotoxin genes.

### 3.4. Antimicrobial Susceptibility Testing

The antimicrobial susceptibility data of the 50 *S. aureus* isolates are shown in Table 5 and Figure 1. These isolates showed the highest resistance rate to ampicillin (56%, 28/50), followed by resistance to erythromycin (54%, 27/50), kanamycin (40%, 20/50), tetracycline (24%, 12/50), sulfamethoxazole-trimethoprim (22%, 11/50), gentamicin (22%, 11/50), ofloxacin (18%, 9/50), oxacillin (14%, 7/50), chloramphenicol (8%, 4/50), and doxycycline (6%, 3/50). All the *S. aureus* isolates were susceptible to vancomycin (0%, 0/50). Moreover, 8 strains (16%) were sensitive to all tested antimicrobial agents, 5 strains (10%) were resistant to one antimicrobial agent, and 8 strains (16%) were resistant to two antimicrobial agents. Beyond expectation, we found that 29 strains (58%) showed MDR phenotype (resistance to three or more types of antimicrobials). Additionally, the 6 strains of MRSA were resistant to ampicillin and oxacillin (100%).

Table 6 and Figure 1 exhibit the relationship between *spa* typing and MRSA. The percentages of MRSA in *spa* types t030 and t4431 strains were 100 and 60%, respectively. The isolates with *spa* types of t3904, t189, t4431, t527, t2844, t267, and t4682 were all non-MRSA strains.

### 3.5. Detection of the Biofilm Formation Capacity of S. aureus

The biofilm-forming abilities of all the 50 *S. aureus* isolates were confirmed by microtiter plate and MicroELISA auto reader assay. As shown in Figure 2A, the positive control *strain* NCTC8325 formed dense biofilm after incubation at 37 °C for 24 h. As shown in Figure 1, 68% (34/50) of the isolates formed strong biofilms, while 32% (16/50) of the isolates formed moderate biofilms. The 50 *S. aureus* isolates from raw milk samples in artisanal dairy retail stores were identified as 9 *spa* types (Table 3 and Figure 1). The biofilm formation abilities of 6 strains of type t3094 were higher than that of NCTC8325, while 5 strains of type t4431, 2 strains of type t2844, and 2 strains of type t4682 were lower than that of NCTC8325. The other *spa*-type strains (t3904, t189, t4431, t030, t527, and t267), compared with NCTC8325, showed no obvious characteristic difference in biofilm formation ability (Figure 2B, Figure 1). These results further confirm that *S. aureus* of different *spa* types isolated from raw milk generally has a strong ability to form biofilm in vitro, which may cause harm to public health security.

## 4. Discussion

Previous studies showed that *S. aureus*, particularly those strains with MDR phenotypes and capacity of producing biofilm and enterotoxins, might contaminate raw milk and dairy products, which may cause an extremely grave public health issue [17,39,40,41]. In the present investigation, we conducted 10-month monitoring of handmade dairy retail stores in Hefei, China to evaluate the antibiotics resistance, virulence, and biofilm formation of *S. aureus* isolates in raw milk.

In our research, 72.5% (50/69) of raw milk samples were positive for *S. aureus* during 10 months of monitoring. The data were consistent with several previous reports, which demonstrated that the detection rate of *S. aureus* in raw milk was 66.7% in Malaysia [42], 77.4% in southern Xinjiang, China [27], and 83% in Italy [43]. On the contrary, comparing the detection rate of *S. aureus* in raw milk (27.7%) and that of ready-to-eat (RTE) food (12.5%) in some areas of China, our results show that the detection rate is higher [44,45]. Overall, it is common to detect *S. aureus* in raw milk that is subsequently processed into fermented yogurt, pasteurized milk, and powder. The reasons are that there might be inappropriate hygiene conditions in raw milk processing areas in different regions and raw milk might come from cows infected by *S. aureus*. Additionally, through *spa* typing, the statistics emphasized the genetic diversity of *S. aureus* isolates from raw milk. The 50 *S. aureus* isolates from the raw milk of artisanal dairy retail stores were grouped into 9 *spa* types. Among them, four *spa* types (t189, t034, t030, and t267), which have been repeatedly reported as isolated *S. aureus* in dairy farms, hospitals, and foods in China, were also identified in these isolates [5,28,29].

The results of the antimicrobial susceptibility test indicated that more than half of the *S. aureus* isolates were resistant to ampicillin and erythromycin. This result was not surprising, because β-lactams and macrolides were widely prescribed to treat bovine mastitis caused by *Staphylococcus* and *Streptococcus*/*Enterococcus* [46,47]. Previous studies performed in China revealed that the prevalence of erythromycin resistance was 58.7% in Shandong, 44.6% in southern Xinjiang, and 46.3% in northern areas [27,30,44], which is similar to our data. However, compared with the erythromycin resistance rate of *S. aureus* isolates from retail food in Beijing, our results were significantly higher [48]. For kanamycin resistance, our results were higher than isolates of *S. aureus* from raw milk as well as dairy products in other areas of China [44,49]. It was shown that 74% of the isolates showed resistance to two or more antibiotics, and 58% of the *S. aureus* isolates were MDR, which was consistent with another study [44]. However, other researchers claimed lower rates of *S. aureus* of MDR [50,51]. In recent years, the emergence of MDR *S. aureus*, especially MRSA, has become an increasingly serious public health concern [52,53]. In our data, six *S. aureus* isolates containing *mecA* were identified as MRSA strains (12%), which was higher than that observed (0.9%) in RTE foods from Shanxi Province, China [54]. By contrast, several studies have shown that the detection rates of MRSA isolated from raw milk and relevant products are similar to our data [49,55]. This increasing prevalence of *S. aureus* observed in this study may be due to antibiotics abuse and other factors that have led to the emergence of MDR *S. aureus*. Consequently, the prevalence of MDR *S. aureus* and MRSA from raw milk used to prepare pasteurized milk and fermented yogurt, as well as the spread of antibiotic-resistant strains, may represent a potential hazard to consumers.

In China, SFP was the third most common bacterial disease from 2011 to 2016, after *Vibrio parahaemolyticus* and *Salmonella* [56]. It has been confirmed that SFP triggered by *S. aureus* is related to the expression of SEs. The discovery of enterotoxin genes in *S. aureus* isolated from food in different regions is thought to be common [5,57,58]. Therefore, this study assessed the presence of genes encoding SEs in all the 50 *S. aureus* isolates. Results data indicated that 24% of all isolates harbored one or more genes encoding *selp*, *sea*, *sec*, *sei*, *ser*, *selj*, and *seh*, and 6% of isolates contained three enterotoxin genes. In other regions of China, the percentage of *S. aureus* isolates from raw milk or dairy products carrying SEs genes is higher than that in this study [59,60,61,62]. Additionally, *sea* has been widely considered the most common reason for SFP globally [63,64]. For instance, the *sea* was the enterotoxin gene with the highest detection rate in clinical isolates of *S. aureus* involved in food poisoning events in China [17]. The *sea* gene detected in this study was consistent with previous studies. However, the dominance of *selp* observed in the present study was not consistent with previous findings. The prevalence difference of genes encoding SEs in *S. aureus* may be due to the different geographical locations of these strains. Overall, *S. aureus* isolated from raw milk used to prepare pasteurized milk and fermented yogurt carried few SEs genes, which is optimistic.

The biofilm formation ability of *S. aureus* has been increasingly recognized as a significant virulence trait [65]. A previous study demonstrated that bacteria form biofilm on the surface of dairy processing equipment; thus, the organisms inside the biofilm might be more able to withstand temperature and pH changes than planktonic organisms [66]. A subsequent study tried to establish an association between *S. aureus* genotype, *spa* type, and biofilm formation ability [38]. For this reason, this study also investigated the relationship between the ability of in vitro biofilm formation and *spa* typing of all *S. aureus* isolates. We found that all 50 *S. aureus* from raw milk could form biofilm, although at different intensities, and these results agreed with two previous investigations conducted in Beijing and Xingjiang, China [27,36]. On the contrary, a study conducted in Brazil showed that approximately 45% of *S. aureus* strains isolated from raw milk had the capacity for biofilm formation [67]. Simultaneously, our data indicated that there was a failure of a specific relationship between the ability of biofilm formation and the type of *spa*, which was consistent with the research results from E. Thiran et al. [38]. The reasons may be that staphylococcal protein A is a vital virulence factor of *S. aureus*, which plays a role in proteinaceous biofilm formation and is highly conserved [68,69], but the biofilm formation of *S. aureus* was regulated by multiple genes (such as SigB factor). A previous study showed that a point mutation (Q225P) of SigB promoted the formation of biofilm [70]. Therefore, there might not be a definite relationship between the *spa* type and biofilm formation ability. In short, the high prevalence of biofilm formation in *S. aureus* isolates demonstrates the necessity for artisanal dairy retailers to refine their quality assurance systems to reduce and eliminate these strains.

## 5. Conclusions

The monitoring of the antibiotics resistance, virulence, and biofilm formation of *S. aureus* in raw milk from artisanal dairy retail stores was conducted in the present study. The present investigation reveals that the detection rate of *S. aureus* in raw milk was 72.5% (50/69), and 58% (29/50) and 12% (6/50) isolates exhibited MDR and MRSA phenotypes, respectively. Furthermore, the high positive rate of biofilm formation and low detection rate of SEs genes were the main characteristics of these isolates. Considering its clinical significance, this study suggests that raw milk as a possible transmission route of *S. aureus* cannot be neglected. To prevent the spread of *S. aureus*, effective measures should be taken during the processing of raw milk to ensure the safety of relevant products.

## Figures and Tables

**Figure 1 foods-11-02185-f001:**
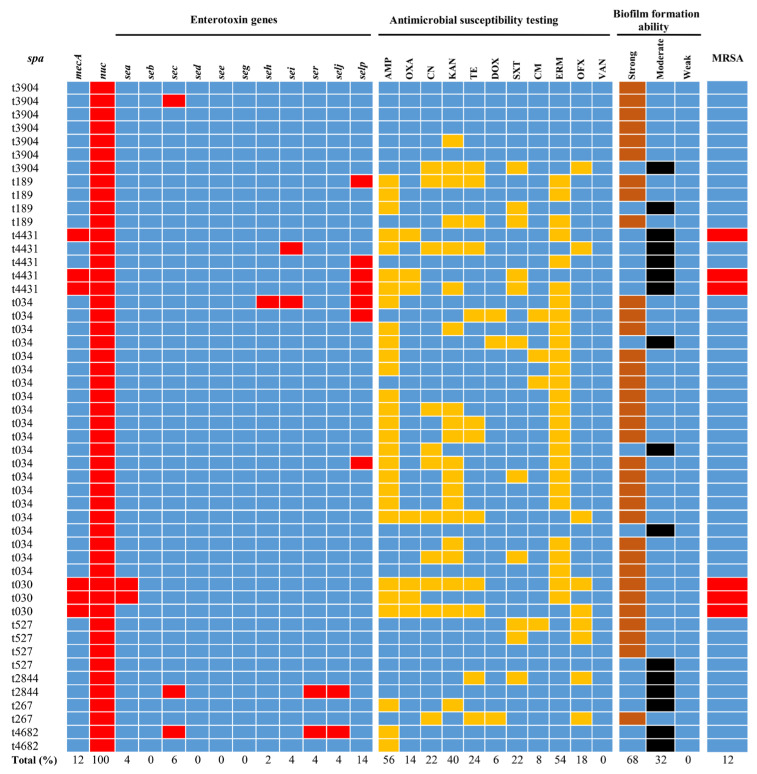
Antimicrobial susceptibility testing (AST), enterotoxin genes, *mecA* gene, biofilm formation, and molecular characterization of 50 *S. aureus* isolates from raw milk in Hefei, China. Fifty isolates were grouped into nine *spa* types. The results of AST are shown in different colors according to isolates’ diameter of inhibition zone in response to different antimicrobial agents. Blue squares indicate susceptibility, yellow squares indicate resistance. The detection of enterotoxin genes and *mecA* genes is summarized on a heat map. Red squares denote that the studied genes were detected in those isolates. Blue squares denote that those isolates lack the studied genes. The ability of isolates to form biofilms is shown in different colors. Brown squares represent strong biofilm isolates formed. Black squares represent moderate biofilm isolates formed. Antimicrobial agents used are abbreviated as follows: AMP = ampicillin; OXA = oxacillin; CN = gentamicin; KAN = kanamycin; TE = tetracycline; DOX = doxycycline; SXT = sulphamethoxazole-trimethoprim; CM = chloramphenicol; ERM = erythromycin; OFX = ofloxacin; VAN = vancomycin. All isolates were tested for antimicrobial susceptibility according to the guidelines of the CLSI.

**Figure 2 foods-11-02185-f002:**
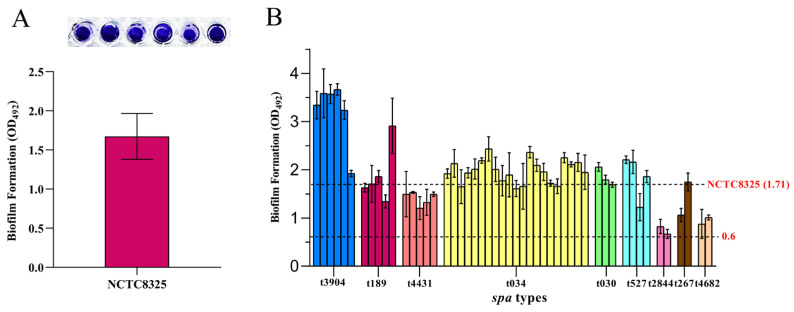
Biofilm formation ability of *S. aureus* NCTC8325 and 50 isolates. (**A**) As the positive control, detection of biofilm formation ability of *S. aureus* NCTC8325 by microtiter plate and a MicroELISA autoreader at a wavelength of 492 nm in single wavelength mode. (**B**) Detection of biofilm formation ability of 50 isolates by a MicroELISA autoreader at a wavelength of 492 nm in a single wavelength mode. Error bars indicate SD. The results represent the means of three independent experiments.

**Table 1 foods-11-02185-t001:** Oligonucleotide primers used in this study.

Gene	Primer	Primer Sequence (5′–3′)	Product Size (bp)	Reference or Source
*16S*	27-F	AGAGTTTGATCCTGGCTCAG	1510	This study
	1492-R	TACCTTGTTACGACTT		
*nuc*	SAnuc-F	AGTATATAGTGCAACTTCAAC	448	This study
	SAnuc-R	ATCAGCGTTGTCTTCGCTCCAA		
*mecA*	mecA-F	GTTGTAGTTGTCGGGTTT	445	This study
	mecA-R	CCACATTGTTTCGGTCTA		
*spa*	spa-1113F spa-1514R	TAAAGACGATCCTTCGGTGAGC CAGCAGTAGTGCCGTTTGCTT	*variable*	Ridom
*sea*	GSEAR-1	GGTTATCAATGTGCGGGTGG	102	[15]
	GSEAR-2	CGGCACTTTTTTCTCTTCGG		
*seb*	GSEBR-1	GTATGGTGGTGTAACTGAGC	164	[15]
	GSEBR-2	CCAAATAGTGACGAGTTAGG		
*sec*	GSECR-1	AGATGAAGTAGTTGATGTGTATGG	451	[15]
	GSECR-2	CACACTTTTAGAATCAACCG		
*sed*	GSEDR-1	CCAATAATAGGAGAAAATAAAAG	278	[15]
	GSEDR-2	ATTGGTATTTTTTTTCGTTC		
*see*	SA-U	TGTATGTATGGAGGTGTAAC	213	[15]
	SA-E rev	GCCAAAGCTGTCTGAG		
*seg*	SEG-F	GTTAGAGGAGGTTTTATG	198	[15]
	SEG-R	TTCCTTCAACAGGTGGAGA		
*seh*	SEH-F	CAACTGCTGATTTAGCTCAG	173	[15]
	SEH-R	CCCAAACATTAGCACCA		
*sei*	SEI-F	GGCCACTTTATCAGGACA	328	[15]
	SEI-R	AACTTACAGGCAGTCCA		
*ser*	SER 1	AGATGTGTTTGGAATACCCTAT	123	[15]
	SER 2	CTATCAGCTGTGGAGTGCAT		
*selj*	SEJ-F	GTTCTGGTGGTAAACCA	131	[15]
	SEJ-R	GCGGAACAACAGTTCTGA		
*selp*	SEP-F	TCAAAAGACACCGCCAA	396	[15]
	SEP-R	ATTGTCCTTGAGCACCA		

**Table 2 foods-11-02185-t002:** Prevalence of *S. aureus* in raw milk of artisanal dairy retail stores in Hefei, China.

Monitoring Period (Month)	No. of Samples	No. of MRSA ^1^ Isolates	No. of Non-MRSA Isolates	No. and Proportion of Positive Samples of *S. aureus*
10	69	6	44	50 (72.5%)

^1^ MRSA = methicillin-resistant *S. aureus*.

**Table 3 foods-11-02185-t003:** *spa* types of the isolated *S. aureus*.

*spa* Type	*spa* Repeat Succession	No. and Proportion of Isolates
t3904	07-23-12-21-17-34-34-34-34	7 (14%)
t189	07-23-12-21-17-34	4 (8%)
t4431	07-12-21-17-13-34-33-13	5 (10%)
t034	08-16-02-25-02-25-34-24-25	21 (42%)
t030	15-12-16-02-24-24	3 (6%)
t527	07-23-12-21-17-34-34-34-34-34-33-34	4 (8%)
t2844	07-16-34-33-34	2 (4%)
t267	07-23-12-21-17-34-34-34-33-34	2 (4%)
t4682	26-34-34-34-33-34	2 (4%)

**Table 4 foods-11-02185-t004:** Distribution of enterotoxin genes.

Enterotoxin Genes	Isolate Code No.	Detection Rate
*sea*	49, 50	4%
*seb*	/	0
*sec*	2, 46, 47	6%
*sed*	/	0
*see*	/	0
*seg*	/	0
*sei*	7, 26	4%
*seh*	7	2%
*ser*	46, 47	4%
*selj*	46, 47	4%
*selp*	7, 8, 9, 19, 34, 35, 42	14%

**Table 5 foods-11-02185-t005:** Antimicrobial susceptibility of the study isolates to the 11 antimicrobial agents.

Antibiotic Class	Antimicrobial	No. and Proportion of Resistant Isolates
β-Lactams	Ampicillin	28 (56%)
	Oxacillin	7 (14%)
Aminoglycosides	Gentamicin	11 (22%)
	Kanamycin	20 (40%)
Tetracyclines	Tetracycline	12 (24%)
	Doxycycline	3 (6%)
Sulfonamides	Sulfamethoxazole-trimethoprim	11 (22%)
Chloramphenicol	Chloramphenicol	4 (8%)
Glycopeptides	Vancomycin	0 (0%)
Macrolides	Erythromycin	27 (54%)
Quinolones	Ofloxacin	9 (18%)
No resistance to an antimicrobial agent		8 (16%)
Resistant to 1 antimicrobial agent		5 (10%)
Resistant to 2 antimicrobial agents		8 (16%)
Multi-drug resistant		29 (58%)

**Table 6 foods-11-02185-t006:** Relationship between *spa* typing and MRSA.

*spa* Type (No)	**No. of Isolates**	**No. and Proportion of Positive Samples of MRSA**
t3904	7	0 (0%)
t189	4	0 (0%)
t4431	5	3 (60%)
t034	21	0 (0%)
t030	3	3 (100%)
t527	4	0 (0%)
t2844	2	0 (0%)
t267	2	0 (0%)
t4682	2	0 (0%)
Total	50	6 (12%)

## Data Availability

The data presented in this study are available on request from the corresponding author.

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
