# Peer review of "Antibiotics Resistance and Virulence of Staphylococcus aureus Isolates Isolated from Raw Milk from Handmade Dairy Retail Stores in Hefei City, China"

_foods, 2022, doi:10.3390/foods11152185_

Round 1
Reviewer 1 Report
The present manuscript describes “Antibiotics resistance and virulence of Staphylococcus aureus isolates isolated from raw milk from handmade dairy retail store in Hefei city, China”. The authors collected a total of 69 different raw milk samples from handmade dairy products retail store in Hefei, Anhui Province, China. The authors investigated the antimicrobial susceptibility, virulence, biofilm formation, and genetic diversity of S. aureus in raw milk. The resistance rates of these isolates to ampicillin, erythromycin, kanamycin, tetracycline, sulfamethoxazole-trimethoprim, gentamicin, ofloxacin, oxacillin, chloramphenicol, and doxycycline were 56%, 54%, 40%, 24%, 22%, 22%, 18%, 14%, 20 8%, and 6%, respectively. The present investigation revealed that the detection rate of S. aureus in raw milk was 72.5% (50/69), and 58% (29/50) and 12% (6/50) isolates exhibited MDR and MRSA phenotypes, respectively. Furthermore, the high positive rate of biofilm formation and low detection rate of SEs genes were the main characteristics of these isolates. Considering its clinical significance, this study suggested that raw milk as a possible transmission route of S. aureus cannot be neglected. To prevent the spread of S. aureus, effective measures should be taken during the processing of raw milk to ensure the safety of food products.
Author Response
The present manuscript describes “Antibiotics resistance and virulence of Staphylococcus aureus isolates isolated from raw milk from handmade dairy retail store in Hefei city, China”. The authors collected a total of 69 different raw milk samples from handmade dairy products retail store in Hefei, Anhui Province, China. The authors investigated the antimicrobial susceptibility, virulence, biofilm formation, and genetic diversity of S. aureus in raw milk. The resistance rates of these isolates to ampicillin, erythromycin, kanamycin, tetracycline, sulfamethoxazole-trimethoprim, gentamicin, ofloxacin, oxacillin, chloramphenicol, and doxycycline were 56%, 54%, 40%, 24%, 22%, 22%, 18%, 14%, 20 8%, and 6%, respectively. The present investigation revealed that the detection rate of S. aureus in raw milk was 72.5% (50/69), and 58% (29/50) and 12% (6/50) isolates exhibited MDR and MRSA phenotypes, respectively. Furthermore, the high positive rate of biofilm formation and low detection rate of SEs genes were the main characteristics of these isolates. Considering its clinical significance, this study suggested that raw milk as a possible transmission route of S. aureus cannot be neglected. To prevent the spread of S. aureus, effective measures should be taken during the processing of raw milk to ensure the safety of food products.
Reply: Thank you for your recognition of the importance of our work, and we appreciate your very insightful comments.
Reviewer 2 Report
The authors present a study about the virulence and antibiotic resistance of Staphylococcus aureus from raw milk used in handmade dairy products.
The article is well written, and the results are very interesting and can be helpful for the industry.
There are, however, to be solved:
Lines 36 and 37: This sentence is very general, stating the number of estimated cases of SFP each year and then showing a precise number of hospitalisations and deaths. To which year are these deaths correspond?
Line 102: TTable 1 states that the authors present the primers used in their study. However, they also show the primers used in other studies. Did the authors use all these primers? If not, separate this information.
Lines 117 to 119: How did the authors choose the quantity of antibiotics to use in the antimicrobial susceptibility study?
Figure 2 is cited in the text before figure 1. Also, please put figures 1 and 2 closer to the text the first time they are mentioned.
Is it common to find S. aureus in raw milk after cheese production? Are there reports about findings of this bacteria in the final products? Why did the authors not also analyse the final products available for the consumers?
Author Response
- Lines 36 and 37: This sentence is very general, stating the number of estimated cases of SFP each year and then showing a precise number of hospitalisations and deaths. To which year are these deaths correspond?
Reply: We apologized for the inaccurate description. We have modified this sentence and added a definite time. References are also replaced with articles with original data. (Lines 37-39)
- Line 102: TTable 1 states that the authors present the primers used in their study. However, they also show the primers used in other studies. Did the authors use all these primers? If not, separate this information.
Reply: Thanks for your comments. Indeed, we used all listed primers in this study. This is explained in lines 111-112 of the manuscript.
- Lines 117 to 119: How did the authors choose the quantity of antibiotics to use in the antimicrobial susceptibility study?
Reply: Thanks for your comments. In this study, the quantity of antibiotics was selected according to the guidelines of the Clinical and Laboratory Standards Institute (CLSI, 2015), as explained in lines 120-121 of the manuscript.
- Figure 2 is cited in the text before figure 1. Also, please put figures 1 and 2 closer to the text the first time they are mentioned.
Reply: We are quite in favor of your suggestion. According to your suggestion, we have put figures 1 and 2 in the manuscript in the position where they were mentioned for the first time.
- Is it common to find S. aureus in raw milk after cheese production? Are there reports about findings of this bacteria in the final products? Why did the authors not also analyze the final products available for the consumers?
Reply: Some published studies have shown that detection of S. aureus from cheese is very common1-5. Some studies have reported that the detection of S. aureus in final products (such as: pasteurized milk and milk powder) from raw milk6-10. However, due to the different sources of raw milk, different food microbial detection standards, and the different processing conditions in different regions, the detection rates in different areas are different.
Although the case of S. aureus residues has also been reported in final products, the detection rate is far fewer than that in raw milk. In addition, numerous articles have been published on the prevalence and characterization of S. aureus in the final products available for the consumers, so this study did not detect and analyze the final products. However, there are other researchers in our research group who are engaged in the analysis of the prevalence and characterization of pathogenic bacteria in final products.
References:
- Johler, S et al. “Tracing and inhibiting growth of Staphylococcus aureus in barbecue cheese production after product recall.” Journal of dairy science vol. 99,5 (2016): 3345-3350. doi:10.3168/jds.2015-10689
- Cai, H. , et al. "Prevalence and characteristics of Staphylococcus aureus isolated from Kazak cheese in Xinjiang, China." Food Control 123.12(2020):107759. doi:10.1016/j.foodcont.2020.107759
- Aragão, B B et al. “Short communication: High frequency of β-lactam-resistant Staphylococcus aureus in artisanal coalho cheese made from goat milk produced in northeastern Brazil.” Journal of dairy science vol. 102,8 (2019): 6923-6927. doi:10.3168/jds.2018-16162
- Johler, S et al. “Short communication: Characterization of Staphylococcus aureus isolated along the raw milk cheese production process in artisan dairies in Italy.” Journal of dairy science vol. 101,4 (2018): 2915-2920. doi:10.3168/jds.2017-13815
- Alghizzi, Mashael, and Ashwag Shami. “The prevalence of Staphylococcus aureus and methicillin resistant Staphylococcus aureus in milk and dairy products in Riyadh, Saudi Arabia.” Saudi journal of biological sciences vol. 28,12 (2021): 7098-7104. doi:10.1016/j.sjbs.2021.08.004
- Dai, Jingsha et al. “Prevalence and Characterization of Staphylococcus aureus Isolated From Pasteurized Milk in China.” Frontiers in microbiology vol. 10 641. 2 Apr. 2019, doi:10.3389/fmicb.2019.00641
- Yehia, Hany M et al. “Prevalence of methicillin-resistant (mecA gene) and heat-resistant Staphylococcus aureus strains in pasteurized camel milk.” Journal of dairy science vol. 103,7 (2020): 5947-5963. doi:10.3168/jds.2019-17631
- Liu, Siyuan et al. “Faster Detection of Staphylococcus aureus in Milk and Milk Powder by Flow Cytometry.” Foodborne pathogens and disease vol. 18,5 (2021): 346-353. doi:10.1089/fpd.2020.2894
- Wang, Xin et al. “Characterization of Staphylococcus aureus isolated from powdered infant formula milk and infant rice cereal in China.” International journal of food microbiology vol. 153,1-2 (2012): 142-7. doi:10.1016/j.ijfoodmicro.2011.10.030
- Ibrahim, Aml S et al. “Safety and quality aspects of whole and skimmed milk powders.” Acta scientiarum polonorum. Technologia alimentaria vol. 20,2 (2021): 165-177. doi:10.17306/J.AFS.0874
Reviewer 3 Report
LINE 116: repetition of the sentence “all isolate of S. aureus were tested for antimicrobial susceptibility by disk diffusion”.
LINE 117: According to CLSI 2021, oxacillin disk testing is not reliable for S. aureus. I suggest to re-test susceptibility to oxacillin by broth microdilution method, as performed for vancomycin, or using cefoxitin disks as surrogate as recommended by CLSI.
LINES 140-141: Did you classify the biofilm formation basing on previous publications? If yes, please report the reference.
LINE 142, PARAGRAPH 2.6: please, specify the DNA isolation method
FIGURE 2: even if the interpretative criteria are reported in the text, it is my opinion that the addition of a graphic legend next to the graph would make the interpretation of the results immediate.
Author Response
- LINE 116: repetition of the sentence “all isolate of S. aureus were tested for antimicrobial susceptibility by disk diffusion”.
Reply: We apologized for the repeated escription. According to your suggestion, we have deleted this repeated sentence in the revised manuscript in line 123.
- LINE 117: According to CLSI 2021, oxacillin disk testing is not reliable for S. aureus. I suggest to re-test susceptibility to oxacillin by broth microdilution method, as performed for vancomycin, or using cefoxitin disks as surrogate as recommended by CLSI.
Reply: We are quite in favor of your suggestion. As you suggested, we retested the sensitivity to oxacillin with the broth microdilution method and modified it in the manuscript. (Lines 120 and 130)
- LINES 140-141: Did you classify the biofilm formation basing on previous publications? If yes, please report the reference.
Reply: Thanks for your comments. Indeed, the classification of biofilms is based on previous studies1-3. We added relevant references to the revised manuscript. (Lines 148-149)
- Wang, Wei et al. “Prevalence and Characterization of Staphylococcus aureus Cultured From Raw Milk Taken From Dairy Cows With Mastitis in Beijing, China.” Frontiers in microbiology vol. 9 1123. 22 Jun. 2018, doi:10.3389/fmicb.2018.01123
- Ren, Qiang et al. “Prevalence and characterization of Staphylococcus aureus isolates from subclinical bovine mastitis in southern Xinjiang, China.” Journal of dairy science vol. 103,4 (2020): 3368-3380. doi:10.3168/jds.2019-17420
- Díez-García, Miryam et al. “Influence of serotype on the growth kinetics and the ability to form biofilms of Salmonella isolates from poultry.” Food microbiology vol. 31,2 (2012): 173-80. doi:10.1016/j.fm.2012.03.012
- LINE 142, PARAGRAPH 2.6: please, specify the DNA isolation method.
Reply: We apologized for the imperfect description. Methods for DNA extraction have been added to the revised manuscript. (Line 160)
- FIGURE 2: even if the interpretative criteria are reported in the text, it is my opinion that the addition of a graphic legend next to the graph would make the interpretation of the results immediate.
Reply: According to your suggestion, we have added interpretation criteria in the revised manuscript. (Lines182-193)
Round 2
Reviewer 2 Report
The authors revised the text according to the reviewer's comments and the final document is greatly improved. The article can be accepted as it is.